# Ambient Temperature and Associations with Daily Visits to a Psychiatric Emergency Unit in Sweden

**DOI:** 10.3390/ijerph16020286

**Published:** 2019-01-21

**Authors:** Hanne Krage Carlsen, Anna Oudin, Steinn Steingrimsson, Daniel Oudin Åström

**Affiliations:** 1Psykiatri Affektiva, Sahlgrenska University Hospital, 416 50 Gothenburg, Sweden; steinn.steingrimsson@vgregion.se; 2Occupational and Environmental Medicine, Gothenburg University, 40530 Gothenburg, Sweden; 3Occupational and Environmental Medicine, Lund University, 223 63 Lund, Sweden; anna.oudin@umu.se; 4Section of Sustainable Health, Department of Public Health and Clinical Medicine, Umeå University, 901 87 Umeå, Sweden; daniel.oudin.astrom@umu.se; 5Department of Psychiatry and Neurochemistry, Institute of Neuroscience and Physiology, Sahlgrenska Academy, University of Gothenburg, 405 30 Gothenburg, Sweden

**Keywords:** psychiatric disorders, mental illness, environmental epidemiology, climate

## Abstract

High or low ambient temperatures pose a risk factor for the worsening or onset of psychiatric disorders. The aim of this study was to investigate the association between ambient temperature and psychiatric emergency visits in an urban region in a temperate climate. The daily number of visits to a psychiatric emergency room (PEVs) at Sahlgrenska University Hospital, Gothenburg, Sweden and the daily mean temperature were extracted for the study period 1 July 2012 to 31 December 2017. Case-crossover analysis with distributed lag non-linear models was used to analyse the data by season. The warm season was defined as May to August and the cold season as November to February. Shorter lags periods were used for the warm season than the cold season. In the analysis, temperatures at the 95th percentile was associated with 14% (95% confidence interval (CI): 2%, 28%) increase in PEVs at lag 0–3 and 22% (95%CI: 6%, 40%) for lags 0–14 during the warm season, relative to the seasonal minimum effect temperature (MET). During the cold season temperatures at the 5th percentile were associated with 25% (95% CI: −8%, 13%) and 18% (95% CI: −30%, 98%) increase in PEVs at lags 0–14 and 0–21 respectively. We observed an increased number of PEVs at high and low temperatures; however, not to a statistically significant extent for low temperatures. Our findings are similar to what has been found for somatic diseases and in studies of other mental health outcomes in regions with more extreme climates. This merits the inclusion of individuals with psychiatric disorders in awareness planning for climate warning systems.

## 1. Introduction

Mortality and morbidity follow a seasonal pattern [1]. Seasonal patterns have been observed for completed suicides in many parts of the world with peaks in spring and early summer [2]. Schizophrenia [3] has been reported to follow a seasonal pattern, but there is a lack of evidence regarding seasonality patterns for admissions due to schizoaffective disorders and psychosis [3,4]. Recently other mental health outcomes have been shown to be associated with high ambient temperatures, such as hospital admissions due to any mental disorder [5], and higher rates of hospital admissions due to mania [6,7], and involuntary psychiatric admissions [8] have been reported during warmer and sunnier months. Much of the literature about mental health and temperature pertain to heat waves, and admissions for mental and behavioural disorders increase during heat waves, more so in the elderly [9]. The increase in the number of mental hospital admissions during heat waves is apparent also when stratifying by diagnoses such as organic mental illnesses; dementia; mood (affective) disorders; neurotic, stress-related, and somatoform disorders; disorders of psychological development; and senility, as well as anxiety, behavioural and personality disorders [10,11]. Furthermore, mental and behavioural emergency department visits increase during heat waves for schizophrenia and mood disorders [12].

Relatively few studies have reported mental health outcomes in the cold season [4]. Associations with cold extremes have been observed for hospital admissions for dementia, senility and neurotic disorders [10,13], whereas another study observed no association [12]. Lower winter temperatures were associated with increased mortality risks in individuals younger than 65 years with a history of mental illness or drug abuse in a Swedish register study [14].

Considering global climate change, understanding the effects of climate on human mental health is of great interest to public health [15]. Heat and mental health are not well-studied in colder climates, and mental health issues are a leading cause of illness and costs to society [16]. Determining environmental factors which aggravate mental illness and symptoms of mental illness could help prevention and planning efforts within the health care system in which resources are limited. The aim of the current study was to investigate short-term effects of high and low ambient temperatures on the worsening of mental illness by studying the association between the daily mean temperature and the daily number of psychiatric emergency department visits in a northern climate.

## 2. Materials and Methods

The study setting was Gothenburg municipality (coordinates 57°47’ N, 11°58’ W), on the Swedish west coast. A city of 600,000 inhabitants with a marine west coast climate with relatively mild winters and cool summers, frequent rains and predominant westerly winds and passing low and high pressure systems. The mean annual temperature is 9.8 °C and there is precipitation (more than 0.1 mm) on 150–175 days of the year [17].

The data extraction procedures are presented in Oudin et al. [18]. In brief, daily counts of all psychiatric emergency visits (PEVs) to the emergency department at Sahlgrenska University Hospital, an always-open walk-in public clinic located at Östra Sjukhuset, during the period of 1 July 2012 until 31 December 2017 were extracted. This psychiatric emergency department is the only access point to tertiary psychiatric care in the area although people may seek care at primary care centre for emergencies. The study population was restricted to individuals with a legal address in the Gothenburg municipality.

### 2.1. Exposure Data

Observations of the daily mean temperature were obtained from the Swedish Meteorological and Hydrological Institute data repository (https://opendata-catalog.smhi.se/explore/) from the urban background measuring station Göteborg A for the study period. In the case of missing data (30 days in December 2013), values from the nearest station, a rural station 18 km away, were used.

### 2.2. Outcome Data

The number of daily PEVs varies over time [18] and is related to different reasons for the visit. Most visits are psychiatric emergencies; however, since no referral is needed some might seek for renewel of medication or advice on the care of others. However, the proportion of non-psychiatric reasons is minimal and it is hard to relate other reasons to the exposure variable directly.

### 2.3. Statistical Methods

We analysed the short-term association between daily mean temperature and PEVs using an overdispersed Poisson regression model with a stratum variable for date. Assuming identical exposure across individuals, as is the case in our study, where all individuals are assumed to be exposed to the same temperatures, such Poisson regression models have been shown to yield identical results to those generated from a conditional logistic regression, which is used in a case-crossover study [19,20].

To allow for the effects of temperature on PEVs that are delayed in time as well as non-linear we used a distributed lag non-linear model (DLNM) [21]. Since any effects of heat are assumed to be immediate, we investigated this using lag structures of up to 3 and 14 days. The effects of cold may take longer to emerge, we thus used two different lag structures, of up to 14 and 21 days [22].

A quadratic B-spline and a natural cubic spline were fitted for temperature and time lag respectively. For temperature, we used two equally spaced internal knots and for the shorter time lag one knot was placed on the log-scale and for the longer lags two equally spaced knots were placed on the log-scale. We additionally controlled for Swedish public holidays (Yes/No).

We stratified all analyses by season as the interpretation of percentages is more meaningful in that case, where the warm season was defined as 1 May to 31 August and cold season as 1 November to 28 February. The results are presented for the 5th and 95th percentiles of the seasonal temperature distribution, as relative risks (generated by the Poisson regression) translated into percentage increase in the number of PEVs with the seasonal minimum effect temperature (MET), i.e., the temperature associated with the least PEVs, as a reference. To limit the influence of extreme temperatures on the MET we restricted the MET to between the 5th and 95th percentiles of the seasonal temperature distribution.

Furthermore, we investigated the impact of single extremely hot and cold days (defined as temperatures above and below the 97.5th and 2.5th of the seasonal percentiles respectively).

In addition, we stratified the analyses by sex. These results are not included in the study, due to the short study period and low number of events occurring in each stratum the results were inconclusive (large differences in METs and large confidence intervals), making comparisons between groups difficult. We did not stratify the analyses by diagnoses due to the same reason.

We performed model checks by visual inspection of normally distributed residuals. All analyses were performed with R version 3.4.0 (R Core Team, Vienna, Austria) and we used the DLNM package [23].

## 3. Results

### 3.1. Descriptive

In the warm season, the mean daily temperatures ranged between 5.2 °C and 25.9 °C, with a mean temperature of 16.2 °C. The 5th and 95th percentiles were 9.9 °C and 21.4 °C respectively. In the cold season, the temperatures ranged between −11.8 °C and 13.6 °C, with a mean temperature of 4.3 °C. The 5th and 95th percentile were −8.5 °C and 11.3 °C respectively. Thus, there was a substantial overlap of cold and warm season temperatures.

During the study period, there were 84,230 PEVs, of which 52.8% were by males and 47.2% were by females. The mean PEVs for the study period was 36.3 individuals per day, ranging from 16 to 62 (Figure 1). The mean number of individuals per day for the cold season was 35.6, ranging from 16 to 62, and in the warm season it was 36.9 individuals per day, ranging from 18 to 61.

### 3.2. Analysis

The MET for the warm season was 10.6 °C and 17.4 °C for lags 0–3 and lags 0–14, respectively. Relative to the MET, temperatures at the 95th at lags 0–3 were associated with a statistically significant increase in the number of PEVs of 14% (95% CI: 2%, 28%). At the longer lags, 0–14 days, relative to MET, we found statistically significant increases in the number of PEVs at the 95th percentile of 22% (95% CI: 8%, 40%) respectively (Table 1).

In the cold season, the MET was 7.45 °C, and 6.45 °C for lags 0–14, and lags 0–21, respectively. The number of PEVs increased at the 5th percentile for both lag structures, however, not to a statistically significant extent.

There were no significant associations between RRs for PEVs and cold temperatures in the warm season or warm temperatures in the cold season although such a trend could be observed for both long and short lags in both seasons (Table 1).

On extremely hot days during the warm season (with temperatures above the 97.5th percentile), the number of PEVs increased by 9% (−1%, 20%) and on extremely cold days during the cold season (with temperatures below the 2.5th percentile), the number of PEVs decreased by 7% (95% CI: −16%, 4%).

## 4. Discussion

We found increased psychiatric emergency room visits at temperatures at the 95th percentile in the warm season, for both lags 0–3 and lags 0–14. In the cold season, for lags 0–14 and lags 0–21, temperatures at the 5th percentile were associated with an increased number of PEVs, however, not to a statistically significant extent.

The study setting where the clinic is located is a temperate Swedish west coast climate with relatively warm winters and cool summers, frequent rains and predominant westerly winds. Compared to other study regions in the body of literature in this field Gothenburg has moderate temperatures with the 99th percentile temperature at 21 °C. A study from Shanghai [5] found that mental disorder hospital admissions increased at a threshold of 24.6 °C, which is near the highest temperature observed in the current study. However, a register-based study from Denmark, which has a similar climate to Gothenburg, observed significant correlations between higher temperature and the risk for admissions for manic state as well as by season, meaning that more patients were admitted in a manic state during the summer months [7]. In an Israeli study, the risk for admission due to schizophrenia was increased with a higher monthly temperature, but the temperatures were much higher than in the current study at ranges from 14 to 34 °C [3]. A common feature for these studies is that they assumed a linear association between the exposure and outcome across a temperature range, whereas several studies have shown that temperature health effects often occur at extreme temperatures in a u-shaped curve [5,19]. Incidentally, Page and colleagues observed a near-linear association between ambient temperature and suicide in men and violent suicide [24]. Furthermore, considering the temperature differences between study settings, it is not certain that the biological mechanisms that govern the observed effects are the same, although humans use a variety of physiological, technical, and behavioural adaptation methods to offset climate exposure [25] and it is well-established that optimal temperature (at which the health outcomes are minimised) varies across countries and populations [19].

Although cold weather effects are reported in relatively few studies of mental health outcomes, Peng et al. [5] observed an increased risk for mental disorder admissions in cold weather at longer lags. Furthermore, Orru and Åström reported increased mortality due to assault in cold weather, suggesting an increase in agitation, but the results did not reach statistical significance [26].

Our results provide tentative, albeit non-significant, evidence of an effect during extremely hot summer days, with the number of PEVs increasing with 9% on such days. In previous heat wave studies, temperatures reached or exceeded 35 °C for three or more consecutive days [9,11]. However, the Chicago heat wave studies cite a high heat index [27,28], as high air humidity was a significant factor in the severity of the event; however, the daily maximum temperature still exceeded 35 °C during four days. Other studies have reported thresholds for negative effects on mental or psychosocial emergency department visits at 22.5 °C in Quebec, Canada, [29] and 26.7 °C in Adelaide, Australia, for mental health admissions and mortality in elderly with mental illness [10]. Not only hospital admissions due to mental disorders increase during heat waves, but also those with pre-existing mental illness are a risk factor for dying in heat waves [7,10,13,14,15]. Furthermore, the use of, and non-compliance to, psychotropic drugs such as antidepressants, and anti-Parkinson’s disease medication have been identified as risk factors of heat stroke deaths and heat-related mortality [30,31].

The mechanisms which underlie heat-related increases in mental hospital admissions are not yet fully understood. The major monoamine metabolites of serotonin and dopamine are affected by meteorological factors such as heat [32]. Psychiatric medications such as SSRI, and neuroleptics have an association with heat intolerance [18].

The risk estimates from the current study of mental health outcomes follow a similar trajectory to risk estimates for somatic illness and mortality, with high risks associated with cold temperature occurring at longer lags, and with heat at shorter lags [14,33,34]. The size of the risk estimates found in the current study are moderate compared to those found for heat wave mortality, where individuals with a pre-existing psychiatric illness had OR 3.61 (95% CI, 1.3–9.8) of dying in a heat wave, and OR 1.90 (95% CI, 1.3–2.8) for individuals who took psychotropic medications. However, in those studies, somatic illness most likely accounted for the main effect or mediated the association [30].

An epidemiological study such as the current one cannot prove causation, and we are thus unable to determine if there is a direct physiological effect of high or low temperature on mental illness, or if the observed association is mediated by other factors. Three examples could be; (1), symptoms of underlying or pre-existing somatic disease may worsen in hot or cold temperatures; an example of this is respiratory disease, a leading cause of years of life lost in the population of mentally ill [35], (2) socioeconomic circumstances such as homelessness, (3) weather-associated changes in behaviour, such as spending more time outside when it is warm increasing exposure to heat [36].

A strength of the current study is that the data are based on administrative-register data from a single clinic serving an entire region, minimizing the risk of bias from misclassification. Health care seeking behaviour is unlikely to be biased by socioeconomic status compared to many other countries as fees are minimal. However, the open-door policy of the clinic can mean that not all registered visits are emergencies which may introduce random errors into the data, but is unlikely to bias the results. A further consequence of the different policies in reaching psychiatric care units might mean that the results are difficult to replicate, but, since the outcome is a proxy for worsening in health, the relative increase should be generalizable to populations in areas with similar climate. The exposure in the current study is the ambient temperature measured at a central location. The measuring station does not correctly reflect the exposure of all individuals in the study. However, we assume a similar distribution across the study area, as our interest lay in the change between days with high and low temperatures. We, therefore, consider the exposure measurement adequate for the present study. Due to missing temperature data, we had to replace data with that from a nearby measuring station, but as these data were during a period of moderate temperatures, we do not believe they skewed the results. We did not adjust for daily air pollution concentrations, as it is a mediator in the causal pathway [37]. We have previously shown that increasing levels of air pollutants seem to be another risk factor for increased PEVs [18].

In our study, we did not stratify the number of daily PEVs by diagnoses due to lack of statistical power, but also because the accuracy of diagnoses given in the emergency ward may not be optimal as mental health diagnostics can be a time-consuming process [38]. Following the same argument, we did not stratify on other diagnoses than mental health diagnoses, even though increased rates of admission during heat waves have been reported on a diagnosis-specific level [10,11].

## 5. Conclusions

In this register-based study, we observed statistically significant associations between extreme temperatures and PEVs in a temperate climate setting. Our results should be verified by others, but there is increasing evidence that individuals with psychiatric disorders should be considered a susceptible group requiring special attention in heat warning systems.

## Figures and Tables

**Figure 1 ijerph-16-00286-f001:**
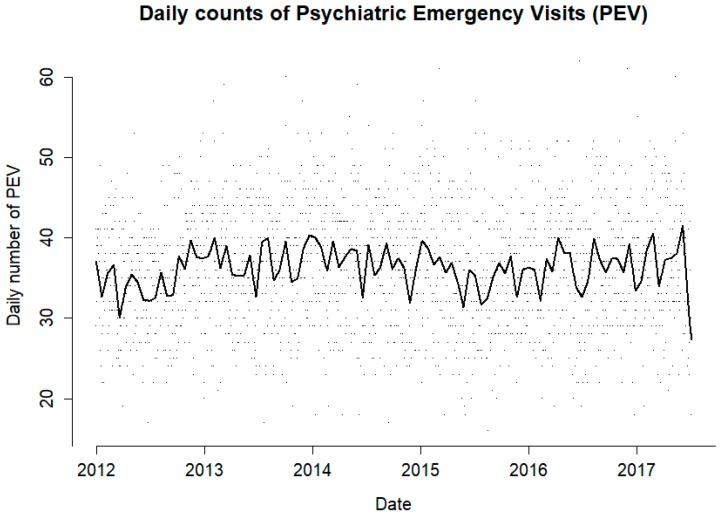
Daily number of individuals visiting the psychiatric emergency room (PEVs) for the duration of the study period (points) and its moving average (line).

**Table 1 ijerph-16-00286-t001:** Effects of temperature presented as relative risks along with 95% confidence intervals on psychiatric emergency visits (PEVs) relative to minimum effect temperature.

Percentiles	Warm Season ^1^		Cold Season ^2^	
	Lags 0–3	Lags 0–14	Lags 0–14	Lags 0–21
5th percentile	1.01 (0.98–1.04)	1.11 (0.94–1.31)	1.25 (0.92–1.71)	1.18 (0.70–1.98)
95th percentile	1.14 (1.02–1.28)	1.22 (1.06–1.40)	1.06 (0.93–1.21)	1.02 (0.79–1.31)

^1^ Warm season defined as the months May to August. ^2^ Cold season defined as the months November to February. Models adjusted for national Swedish holiday.

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
