# Peer review of "Ambient Temperature and Associations with Daily Visits to a Psychiatric Emergency Unit in Sweden"

_ijerph, 2019, doi:10.3390/ijerph16020286_

Round 1

Reviewer 1 Report

The authors have improved the manuscript in relation to the previous version.

Some of the changes in text present errors in the structure of the sentences and I would recommend reviewing them (e.g. "Another pathway hypothesized by Holopainen and colleagues [33] their study of suicide seasonality involves heat generation in brown adipose tissue from noradrenaline signaling after cold exposure").

The references 10 and 11 are cited in both the Introduction and the Discussion with the exact same wording. 

Would it be possible to cite in the Introduction any previous study based on national registries?

While it is justified why air pollution was not included in the study as a covariate, it is unclear why precipitation was and how it relates to the temperature in the given context.

Sex could not be used to stratify the results. However, I wonder whether the effect of sex could be explored by including the effect in the predictions.

It is unclear if the reference to "modifiable environmental factors" is referred to temperature? (line 59), in which case it sounds confusing.

On line 196, the authors state that they restricted MET to be found between the 5th and 95th percentile. Is this correct? As I understand, the results were restricted to those below to the 5th and above the 95th percentile.

Are the results presented on lines 143-145 related to the cold or the warm season? 

I did not notice in the previous version, but the sentence on lines 197-198 does not seem to match the observations of this study - I wonder whether it is incomplete?

On line 220 the possible effect of vacation is acknowledged. This study controlled for public holidays, so I wonder if it is an actual limitation.

The authors should consider presenting the confidence intervals separated with commas instead of hyphens to avoid confusion with negative values in the temperatures. Additionally, 9% (-1% - 20%) on line 144 should indicate it is 95%CI.

Author Response

Response to Reviewer 1 comments.

The authors have improved the manuscript in relation to the previous version.

Author response: Thank you for your thoughtful comments which we believe have improved the quality of the paper.

Some of the changes in text present errors in the structure of the sentences and I would recommend reviewing them (e.g. "Another pathway hypothesized by Holopainen and colleagues [33] their study of suicide seasonality involves heat generation in brown adipose tissue from noradrenaline signaling after cold exposure").

Author response: This sentence was removed from the paper. The entire manuscript is checked for consistency.

The references 10 and 11 are cited in both the Introduction and the Discussion with the exact same wording.

Author response: Thanks for pointing this out to us, we have changed the sentence in the discussion which now reads: “Following the same argument, we did stratify on other diagnoses than mental health diagnoses, even though increased rates of admission during heat waves have been reported on a diagnose specific level (10,11)”

Would it be possible to cite in the Introduction any previous study based on national registries?

Author response: We cite Rocklöv et al (2014) (R55), which used Swedish registers. We added the word register to make this clear to the reader

While it is justified why air pollution was not included in the study as a covariate, it is unclear why precipitation was and how it relates to the temperature in the given context.

Author response: This was added based on the suggestion of reviewer 3. However, the reasoning most likely is the same. Precipitation seems unlikely to cause temperature, and we have thus removed this.

Sex could not be used to stratify the results. However, I wonder whether the effect of sex could be explored by including the effect in the predictions.

Author response: When running a Poisson regression model we must model the daily counts of mortality for men and women separately, resulting in sex specific estimates. We are not entirely sure what is meant by including the effect in the predictions.

It is unclear if the reference to "modifiable environmental factors" is referred to temperature? (line 59), in which case it sounds confusing.

Author response: We agree and have deleted modifiable

On line 196, the authors state that they restricted MET to be found between the 5th and 95th percentile. Is this correct? As I understand, the results were restricted to those below to the 5th and above the 95th percentile.

Author response: We restricted the MET to be found between the 5th and 95th percentiles and report the results of the 5th and the 95th percentiles relative to the MET. We agree that there is ambiguity in the way the methods and the result sections are written and have made changes consistently throughout the revised version to clarify understanding.

Are the results presented on lines 143-145 related to the cold or the warm season?

Author response: Thanks for pointing this out to us. It refers to extremely hot temperatures during the warm season and extremely low temperatures during the cold season. We have added this explanation

I did not notice in the previous version, but the sentence on lines 197-198 does not seem to match the observations of this study - I wonder whether it is incomplete?

Author response: We have deleted this section

On line 220 the possible effect of vacation is acknowledged. This study controlled for public holidays, so I wonder if it is an actual limitation.

Author response: This has now been deleted.

The authors should consider presenting the confidence intervals separated with commas instead of hyphens to avoid confusion with negative values in the temperatures. Additionally, 9% (-1% - 20%) on line 144 should indicate it is 95%CI.

Author response: This is a good suggestion; we have changed this throughout the revised version.

Reviewer 2 Report

While the subject matter of the manuscript is of importance in better understand susceptible populations to high ambient temperatures, some clarifications and improvements are needed.

In the beginning of the introduction you specifically mention heat (line 39), though the manuscript talks specifically about high and low ambient temperatures – may want to clarify by saying ambient temperature here.

Why does the data start in July 2012 and end in December 2017, though the categorizations for warm and cold seasons are May-August and November-February, respectively? Are there some years that have incomplete seasonal profiles (e.g. the first year of analysis and the last)? What impact does this have on the results and seasonality?

Also in Table 1 you define the seasons as Summer and Winter, but in the text you refer to the seasons as ‘warm’ and ‘cold’ – I would stick to one or the other.

I would further clarify in your methods section that Poisson regression model generate relative risk (RR), as this information appears in table 1, but is never defined. Additionally, the reader may require further guidance that an RR of 1.14 indicated a 14% increased risk of PEV hospitalization.

In lines 150-151 of the discussion you end the sentence with “…followed a u-shared curve” – you do not introduce this information as being relevant to temperature related hospitalizations and should be further clarified as it is the first mention and readers may not be aware of what this describes and why.

Line 152, I would indicate a “Swedish west coast climate” as opposed to a west coast climate.

In line 157-161 you related the results of other studies indicating correlation coefficients of 0.12 and 0.14, as well as admissions correlated with temperature by 0.35. This does not make any sense to the reader as we are given no context to what those results entail. Some description of an interpretation and how it reflects the results of your study are warranted.

Lines 182-187 – These are incomplete sentences or “notes to self” and gives great confusion in transitioning to the following paragraph – needs to be fixed.

Lines 213-214 – this is repeated information without any new discussion on the topic of medication.

Line 219 – what is “wards” referring to?

Lines 224-229 – Is this the only hospital in the region? There could be other places that individuals are accessing their healthcare needs and would be a limitation of the study. Also, what do the authors mean by “should be generalizable”? To whom, to which areas? As the authors mentioned before healthcare is accessed different in other countries so generalizability may only apply to temperate regions in Sweden. This needs to be caveated or generalizability needs to be explained in more detail.

Author Response

Response to Reviewer 2's comments.

While the subject matter of the manuscript is of importance in better understand susceptible populations to high ambient temperatures, some clarifications and improvements are needed.

Author response: Thank you for your valuable comments, which again have improved the manuscript.

In the beginning of the introduction you specifically mention heat (line 39), though the manuscript talks specifically about high and low ambient temperatures – may want to clarify by saying ambient temperature here.

Author response: We have made the suggested change.

Why does the data start in July 2012 and end in December 2017, though the categorizations for warm and cold seasons are May-August and November-February, respectively? Are there some years that have incomplete seasonal profiles (e.g. the first year of analysis and the last)? What impact does this have on the results and seasonality?

Author response: Unfortunately, the data collection only started in July 2012. We would naturally have preferred to have data covering a longer time period, but as the time series is rather short we chose to include all relevant information. We do not anticipate this should impact the seasonal analysis as this is defined in the cross-basis of the DLNM model.

Also in Table 1 you define the seasons as Summer and Winter, but in the text you refer to the seasons as ‘warm’ and ‘cold’ – I would stick to one or the other.

Author response: Thanks for pointing this out to us, we are now consistently use warm and cold season

I would further clarify in your methods section that Poisson regression model generate relative risk (RR), as this information appears in table 1, but is never defined. Additionally, the reader may require further guidance that an RR of 1.14 indicated a 14% increased risk of PEV hospitalization.

Author response: This is a good suggestion and we have added this clarification.

In lines 150-151 of the discussion you end the sentence with “…followed a u-shared curve” – you do not introduce this information as being relevant to temperature related hospitalizations and should be further clarified as it is the first mention and readers may not be aware of what this describes and why.

Author response: We agree that this is not relevant and have deleted this sentence.

Line 152, I would indicate a “Swedish west coast climate” as opposed to a west coast climate.

Author response: This has been added.

In line 157-161 you related the results of other studies indicating correlation coefficients of 0.12 and 0.14, as well as admissions correlated with temperature by 0.35. This does not make any sense to the reader as we are given no context to what those results entail. Some description of an interpretation and how it reflects the results of your study are warranted.

Author response: We agree that the correlation coefficients are impossible to interpret without additional information and have removed these and clarified the associations found in these studies in more general terms.

Lines 182-187 – These are incomplete sentences or “notes to self” and gives great confusion in transitioning to the following paragraph – needs to be fixed.

Author response: This is now fixed

Lines 213-214 – this is repeated information without any new discussion on the topic of medication.

Author response: We agree and have removed this sentence and the reference.

Line 219 – what is “wards” referring to?

Author response: This sentence has now been deleted

Lines 224-229 – Is this the only hospital in the region? There could be other places that individuals are accessing their healthcare needs and would be a limitation of the study. Also, what do the authors mean by “should be generalizable”? To whom, to which areas? As the authors mentioned before healthcare is accessed different in other countries so generalizability may only apply to temperate regions in Sweden. This needs to be caveated or generalizability needs to be explained in more detail.

Author response: The hospital whose data we obtained for the study is the only tertiary hospital offering acute psychiatric care in the immediate region, while minor emergencies may be handled by primary centers and we have clarified this in the text (Methods). As for generalizability, we have clarified in the Discussion that we generalize to populations in areas with similar climate as we consider our outcome a proxy for health, regardless of the structure of health care systems.

Reviewer 3 Report

The authors have addressed all my concerns from the original review and the paper has been significantly improved.

Author Response

Response to Reviewer 3's comments.

The authors have addressed all my concerns from the original review and the paper has been significantly improved.

Author response: Thank you for the comments helping us improve the manuscript

Round 2

Reviewer 2 Report

Authors have addressed all comments, manuscript content and clarity improved.